# An Empirical Study of Social Commerce Intention: An Example of China

**Chao-Hsing Lee** [1] and **Chien-Wen Chen** [2,*]

1  School of Economics and Management, Shangrao Normal University, Shangrao 334001, China; jesse5501@gmail.com
2  Department of Business Administration, Feng Chia University, Taichung 40724, Taiwan
*  Correspondence: chencw@fcu.edu.tw; Tel.: +886-4-24517250 (ext. 4621)

**Abstract:** The rise of social networks is rapidly spreading in China. Using social platforms, individuals are no longer just receivers of Internet information, as consumers generate and share contents with others. Social interaction and spontaneous promotion activities are carried out among consumers, but with the growth of traditional e-commerce slowing down, social commerce derived from social networks is gradually taking shape. Based on Hajli's theoretical model, this study uses the social support theory and social commerce construct to study consumers' social commerce behavior from a total of 1277 valid sample questionnaires that were distributed in a social platform environment in China. Through the empirical research evaluation using PLS-SEM, the statistical analysis results prove that social commerce constructs do promote social interaction of consumers. Such constructs have a positive effect on social support and social commerce intentions. In this regard, social support is embodied in information support and emotional support, and has a positive effect on social commerce intention. This study also conducts cross-cultural empirical comparisons. In comparison with Hajli's research, this study has the same results in evaluation of Chinese samples. Among the users who exhibit social commerce intentions, social commerce construction is more important than social support.

**Keywords:** social commerce; e-commerce; social commerce intention

## 1. Introduction

The 43rdChina Statistical Report on Internet Development [1] released by the China Internet Network Information Center (CNNIC) pointed out that as of December 2018, China's online shopping users had reached 610 million, with a utilization rate of 73.6%. Online shopping has clearly become one of the main activities of Internet users, and after years of rapid development, the online consumer market has gradually entered a stage of upgrading. New models, such as social commerce, have continuously enriched consumption scenarios. Consumption upgrading and consumption stratification have also become increasingly prominent, further promoting diversified market development.

Since the launch of Tmall Double 11 Global Shopping Festival in 2009, the annual growth rate has been hitting new highs, from 50 million RMB in the first year to 168.2 billion RMB in 2017 and 213.5 billion RMB in 2018. However, with each year establishing a greater amount of shopping turnover, the rate of growth has slowed down, and traditional e-commerce companies are gradually encountering bottlenecks. Consumers now face higher acquisition costs, and small and medium-sized e-commerce platforms are gradually losing competitiveness. Facing an unclear future, e-commerce giants are seeking a wide range of solutions. They hope to reach new highs in sales, and sales upgrading is the key for doing so. Social commerce has also become the main direction of development in recent years.

Due to the increasing popularity of social networks such as Facebook, Twitter, WeChat, and MicroBlogs in China, people now transmit and exchange information through social networks, branching e-commerce off into social commerce with real and interactive social networks and social relationships on those social networks. Individuals are the content creators on social networking sites. Social commerce makes good use of the functions of social networks to encourage customers to share their personal experiences. The emergence of social commerce has brought great changes to stores and consumers and is the evolution of electronic commerce. Social platforms as carriers help promote commodity sales by means of interactions between stores and consumers, among consumers, or via voluntary promotion by consumers.

Hajli has proposed the social commerce intention model [2], in which social networks promote social support and provide information to guide users on the network to make decisions when consuming. Better social support leads to higher commerce intentions and affects consumers' social behaviors. Therefore, it is important to study social support and the commercial construct of business intentions. But he also mentioned that one of the main limitations of his research is in the sample used. The majority of the participants were from the UK, making the results limited to a particular culture. Testing the model in another region with another group may make comparison possible and have other findings. In social commerce, ethnic culture plays an important role in the content and quality perception of customer reviews [3]. Purchase intention will show distinct effects in different cultures [4]. Cultural characteristics influence people's values and trust [5]. This motivated us to study a comparison of social business intentions in different cultures. This study tries to validate the social commerce intent Model with Chinese samples. Based on Hajli's theoretical model, this study makes certain adjustments to adapt to the actual situation in China. Under China's social network environment, we examine Chinese consumers' rational or emotional social support in social networks and their intentions arising from the social commerce construct and social support when consuming online, explore whether social commerce constructs affect commerce intention and the social support of Chinese consumers, and look at whether social support influences commerce intention.

Social networking sites (SNSs) are a suitable detecting tool in identifying social behavioral [6]. This study was conducted on the platform WeChat. As the popularity of social networks varies from culture to culture, this study mainly focuses on WeChat. With the usage rate of "WeChat Moments" in China at 83% [1], it is undoubtedly the most popular social platform in China and is quite representative of the general consumer population.

## 2. Literature Review

### 2.1. Social Commerce Intention

Social commerce is the evolution of e-commerce. On social platforms, consumers obtain pre-shopping information, communicate with sellers, buy commodities, and share experiences after shopping. Yadav defined social commerce as exchange-related activities that occur in or are affected by personal social networks in the social environment of computers, where the activities correspond to phases of demand identification, pre-purchase, purchase, and later purchase transactions [7]. This definition clarifies the interactive relationship between social commerce members of a social platform network in a computer environment and the phases of consumer decision-making and information exchange [8].

Social commerce is a new trend and refers to the application of social elements such as attention, sharing, communication, discussion, and interaction to the e-commerce transaction process [9]. Interaction and promotion in social commerce are initiated not only by merchants, but by consumers themselves. Consumers' initiative is mainly to share recommended commodities with other consumers or potential users through social platforms [10]. It is the interaction of consumers on social platforms and the demands of e-commerce form social commerce, as well as the fact that such social commerce

has value and sustainability, that make e-commerce giants set social commerce as one of the main strategies for future development.

Various commercial applications based on the Internet that support social interaction and user content use social media to generate social commerce so as to support individuals in their purchase decisions [11]. Social commerce intention is the pre-step of social commerce. The direct result of social commerce intention is social commerce. Therefore, we use social commerce intention to understand consumers' decision-making in social commerce.

## 2.2. Social Support

Social support is a psychological term. It relates to how individuals feel, perceive, or receive care or assistance from others, measures how individuals are cared for, and how they are being helped by people in social groups that respond to them [12]. In the age of the Internet, it was originally thought that people hiding behind keyboards would witness poorer interpersonal relationships. However, with the rise of social networks, social platforms have become an important channel for promoting interpersonal relationships. Research has found that users can obtain social support from social networks [13].

Social support is defined as "social resources" that are available or actually provided to people who are supported by non-professional personnel in the context of groups and informal help relationships [14]. As interaction on social platforms is virtual in nature and often relies on information exchange, such online social support is divided into rational information support and emotional support. Information support refers to providing information, advice, suggestions, or knowledge in a form that may help solve problems and provide practical and specific assistance. Emotional support refers to providing emotional information, care, understanding, or sympathy for the purpose of social and psychological functions [12]. Consumers' exchange of informational and emotional social support significantly facilitates social influence among them [15].

Consumers can support their purchase decisions through social interaction and information acquisition on social networks, which create online social support [16]. The social support discussed in this study refers to Hajli's definition, which refers to individuals obtaining the support of peers in rational or emotional ways through social network platforms.

Therefore, consumers can obtain social support through social networks in a positive way as well as influence their social commerce intentions [17]. Thus, the following is our research assumption.

**Hypothesis (H1).** *Social support has a positive effect on consumers' social commerce intentions.*

## 2.3. Social Commerce Structure

Social commerce constructs (SCC) are social platforms constructed by rating, commenting, using forums and communities, and making and receiving recommendations [18]. E-commerce practitioners use ratings, reviews, and recommendations to develop social commerce or to promote communication or interaction with consumers through third-party online forums and communities [19].

The first important construct of a social commerce platform is to obtain online users' ratings and comments [20]. People can easily publish online assessments and comments on products and services and provide comprehensive information about products and services for potential customers [21]. Online users' comments are regarded as statements of their consumer experiences, and product-related information can be used as a new element of marketing communication. This new mode of communication does not cause any cost to e-commerce practitioners and is regarded as a "free sales assistant" [22]. A second important construct in social commerce covers forums and communities. Members of online forums and communities participate in activities to provide information and form support through interactions among members [23]. The information provided by e-commerce practitioners is based on the perspective of manufacturers and products and is often

considered to be embedded in marketing. Opinions of forums and communities often come from authoritative people or familiar friends and are based on personal experiences or needs, featuring a high degree of trust. At present, the most popular online forums and communities in China include Baidu Tieba, Zhihu, and Douban. The third SCC is recommendation. When online users have no experience in using products or services, they must rely on the experience of other consumers. In the case of WeChat, the most widely used social platform in China, discussions in social groups or recommendations by friends become very important channels. Such experiences and recommendations affect a person's thoughts, feelings, and attitudes, and are likely to play an important role in social commerce intentions. SCCs have empowered consumers through the existence of virtual groups, ratings, reviews, recommendations, and referrals, thereby exhibiting social commerce intentions [24].

The rise of social networks enables consumers to participate in social commerce on social platforms and generate social support, thus affecting their purchase decisions.

**Hypothesis (H2).** *Social commerce constructs have a positive effect on consumers' social commerce intentions.*

**Hypothesis (H3).** *Social commerce constructs have a positive effect on consumer social support.*

The research model is shown in Figure 1 [2].

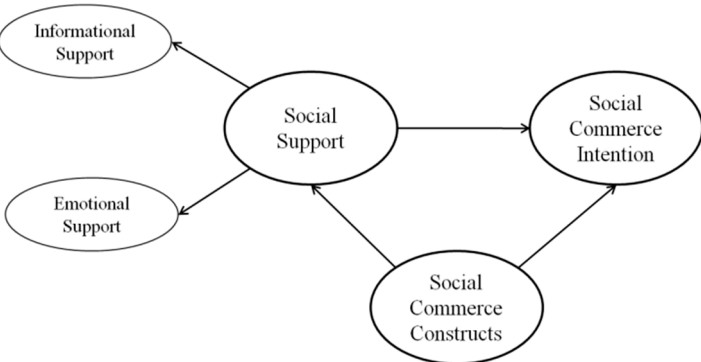

**Figure 1.** Social commerce intention model.

## 3. Research Method

The above research model refers to the theoretical framework of Hajli [2], which is limited to samples from the UK. In order to verify whether the theoretical model can be applied to Chinese society, this study modified the research structure and questionnaire content to adapt to the actual situation in China, and adopted the model of the Chinese Internet community.

The questionnaire was designed with a 7-point Likert scale. Upon completion of the questionnaire design, five target subjects were first asked to fill out the questionnaire, and the semantic understanding of the questionnaire content was tested and revised to improve its readability. After that, another 15 target subjects were invited for pre-test it to confirm the questionnaire quality and response results. Finally, example verification was carried out.

This study conducted the questionnaire survey in April 2019. According to the China Statistical Report on Internet Development, nearly 700 million people in China use WeChat Moments, which is the most popular and representative social platform in China, with a 83% usage rate. Therefore, this study adopted the electronic version of the questionnaire and conducted a questionnaire survey on WeChat. The questionnaire was distributed among WeChat groups, and "red envelope" rewards were given to respondents. One WeChat ID could only fill in one questionnaire to ensure they were answered by valid social platform users. The questionnaire was spread in an effective manner. The questionnaire was regarded as valid if the person who filled in the questionnaire was born between 1960 and 1999 and if the monthly online shopping amount of such person exceeds 50 RMB.

After limiting the above age group and the monthly shopping amount, 1277 valid samples were collected within one week after distributing the questionnaire through WeChat, of which 47% were women and 53% were men. In terms of educational background, 45% were at junior college and senior high school or below, and 55% were undergraduate and postgraduate or above. The samples were evenly distributed in terms of gender and educational background. A total of 35% had a monthly online shopping amount exceeding 500 RMB and 65% had one between 51 and 499 RMB. The monthly online shopping amount was mainly distributed between 200 and 500 RMB.

Although this study refers to Hajli's theoretical framework, WeChat, as the most popular social platform in China, was employed as a platform for the questionnaire survey. In order to avoid cognitive differences, the forums and communities mentioned in the SCCs are also specifically named as Baidu Tieba, Zhihu, Douban, and other websites or apps that are currently popular in China. Although SNSs are appropriate tools in studying social commerce behavior [6], collecting data from WeChat can only represent certain groups. It has limitations in representing only Chinese people. This study uses structural equation modeling (SEM), which has good reliability and effectiveness in path analysis and factor analysis. The partial least square method and structural equation modeling(PLS-SEM) is used to test the hypothesis.

## 4. Research Results

In this study, SmartPLS v.3.28 software was used for statistical analysis. The PLS algorithm was executed to obtain relevant results. The execution results are shown in Figure 2.

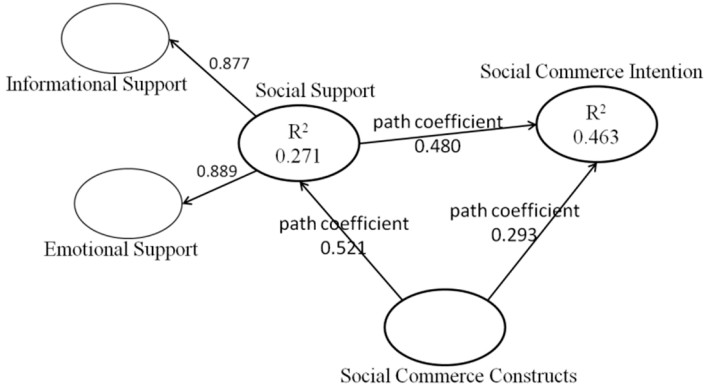

**Figure 2.** Path diagram of social commerce intention.

*4.1. Reliability Analysis*

1.　As can be seen from Figure 2 & Table 1, if the loading of individual factors is greater than 0.5, this mode has internal consistency.

**Table 1.** Factor loading of constructs.

| Codes | Scales | Factor Loading |
|---|---|---|
| | Informational Support | |
| SI1 | On the WeChat, some people would offer suggestions when I needed help. | 0.808 |
| SI2 | When I encountered a problem, some people on the WeChat would give me information to help me overcome the problem. | 0.838 |
| SI3 | When faced with difficulties, some people on the WeChat would help me discover the cause and provide me with suggestions. | 0.793 |
| | Emotional support | |

**Table 1.** *Cont.*

| Codes | Scales | Factor Loading |
|---|---|---|
| SE1 | When faced with difficulties, some people on the WeChat comforted and encouraged me. | 0.796 |
| SE2 | When faced with difficulties, some people on the WeChat listened to me talk about my private feelings. | 0.847 |
| SE3 | When faced with difficulties, some people on the WeChat expressed interest and concern in my well-being. | 0.788 |
| | Social commerce constructs | |
| SCC1 | I will ask my friends on forums and communities to provide me with their suggestions before I go shopping. | 0.791 |
| SCC2 | I am willing to recommend a product that is worth buying to my friends on the WeChat. | 0.761 |
| SCC3 | I am willing to share my own shopping experience with my friends on Baidu Tieba, Zhihu, and Douban or through ratings and reviews. | 0.810 |
| SCC4 | I would like to use people's online recommendations to buy a product. | 0.782 |
| | Social Commerce Intention | |
| IB1 | I am willing to provide my experiences and suggestions when my friends on the WeChat want my advice on buying something. | 0.754 |
| IB2 | I am willing to buy the products recommended by my friends on WeChat. | 0.818 |
| IB3 | I will consider the shopping experiences of my friends on WeChat when I want to shop. | 0.821 |

2. As shown in Table 2, if Cronbach's $\alpha$ values are between 0.70 and 0.98, this model is highly reliable.

**Table 2.** Cronbach's $\alpha$ values and composition reliability.

| | Cronbach's $\alpha$ | Composition Reliability |
|---|---|---|
| Informational Support(IS) | 0.743 | 0.854 |
| Emotional Support(ES) | 0.739 | 0.852 |
| Social Support(SS) | 0.810 | 0.864 |
| Social Commerce Constructs(SCC) | 0.794 | 0.866 |
| Social Commerce Intention(SCI) | 0.717 | 0.840 |

3. According to Table 2, if the composition reliability exceeds 0.7 and reaches about 0.85, the measurement items can reflect the research framework.

*4.2. Validity Analysis*

1. As shown in Table 3, if the average variation extraction amount of each construct is greater than 0.5, the construct has good convergence validity.

**Table 3.** Average variation extraction (AVE).

| | Average Variation Extraction |
|---|---|
| Informational Support | 0.661 |
| Emotional Support | 0.657 |
| Social Support | 0.514 |
| Social Commerce Constructs | 0.618 |
| Social Commerce Intention | 0.638 |

2. Through the cross-load matrix of Table 4, and the square root observation of the average variance extraction, if the correlation degree of all measurement items in the same construct is greater than the correlation coefficient between this construct and other constructs, each construct has good discrimination validity.

**Table 4.** Cross-loading matrix.

|  | **Social Support** | **Social Commerce Constructs** | **Social Commerce Intention** |
|---|---|---|---|
| SE1 | 0.667 | 0.335 | 0.306 |
| SE2 | 0.743 | 0.334 | 0.373 |
| SE3 | 0.720 | 0.392 | 0.386 |
| SI1 | 0.735 | 0.374 | 0.365 |
| SI2 | 0.724 | 0.378 | 0.400 |
| SI3 | 0.709 | 0.423 | 0.495 |
| SCC1 | 0.430 | 0.791 | 0.489 |
| SCC2 | 0.407 | 0.761 | 0.506 |
| SCC3 | 0.392 | 0.810 | 0.495 |
| SCC4 | 0.408 | 0.782 | 0.498 |
| IB1 | 0.456 | 0.387 | 0.754 |
| IB2 | 0.440 | 0.544 | 0.818 |
| IB3 | 0.412 | 0.566 | 0.821 |

### 4.3. ResearchHypothesis and Verification

The path coefficient and the interpretable force (R-square, $R^2$) for the endogenous variables were then estimated by the relationship between the constructs and the question items in the structural model. The path coefficient represents the connection relationship and influence strength between the components, and the causal relationship between the observed variables and the potential variables was hypothetically calculated to verify the research model. Therefore, bootstrapping was executed with SmartPLS and set to be executed 5000 times to obtain relevant results, as shown in Figure 3.

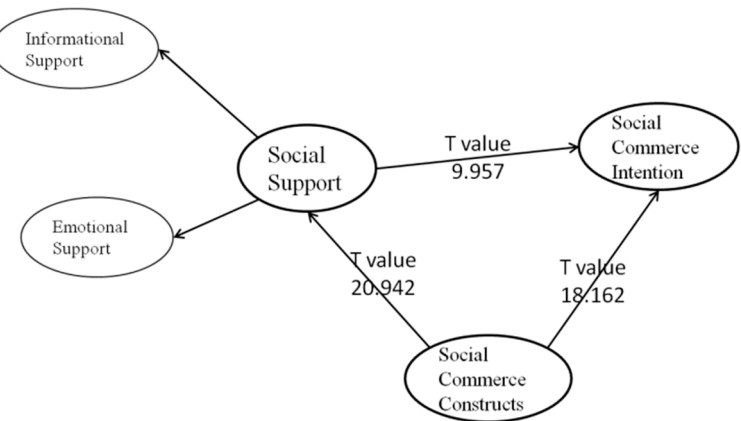

**Figure 3.** Results of bootstrapping implemented by the social commerce intention model.

In this study, the hypothesis between potential variables can be verified through the significance pointer (t-value) in the model, as shown in Table 5.

**Table 5.** Verification of research hypothesis.

|  | Path Factor | t-Value | Decision |
|---|---|---|---|
| H1: Social Support –> Social Commerce Intention | 0.293(***) | 9.957 | supported |
| H2: Social Commerce Constructs –> Social Commerce Intention | 0.480(***) | 18.162 | supported |
| H3: Social Commerce Constructs –> Social Support | 0.521(***) | 20.942 | supported |

* represents $p < 0.1$; ** represents $p < 0.05$; *** represents $p < 0.01$.

Figure 2 shows the path analysis diagram of social support, the social commerce construct, and the social commerce intention model. Social support has a positive effect on consumers' social commerce intentions (H1). The results of this study support this hypothesis, and $R^2$ is 0.463, which indicates that the explanatory power is quite strong. The social commerce construct has a positive effect on consumers' social commerce intentions (H2) and has a positive effect on consumers' social support (H3). The study's results support these hypotheses, and the $R^2$ values are 0.463 and 0.271, respectively, showing strong explanatory power.

Based on Hajli's theoretical model, this study conducted a questionnaire survey on Chinese consumers in the Chinese Internet environment through WeChat, the most popular social network platform in China. Through PLS-SEM statistical analysis, it has been shown that consumers obtain rational or emotional social support through social network platforms, which have a positive effect on personal social commerce intentions. Similarly, social commerce constructs built through ratings, comments, forums, communities, recommendations, and other modes have positive effects on social support and social commerce intentions.

RegardingHajli's research, his study was conducted about Facebook. The survey was carried out in London both in paper questionnaire and online versions. The paper questionnaire was distributed in public areas such as libraries and coffee shops. The author sent 900 questionnaires out and received 230 usable questionnaires, mostly from Londoners. In the research, the author used the PLS-SEM approach.

According to his results, all the paths in his research model were significant. The $R^2$ showed that the model had sufficient explanation power. The path coefficients indicated that social support (0.127) and social commerce constructs (0.5) both had significant effects on social commerce intention. And the direct effect of social commerce constructs on social commerce intention was stronger than that of social support (0.5 > 0.127). This indicated that the social commerce construct is more important than social support in affecting social commerce intention in the model. Also, social commerce constructs have a significant effect on social support (0.479).

We made a comparison to the above empirical research results from Hajli and made a table.

1. As can be seen from Table 6, both studies used the same model, similar questionnaire items, and the same method of hypothesis testing. The empirical approaches were similar but somewhat different. This study has modified some questionnaire items to fit the actual situation in China. The SNS selected in this study was WeChat, and data was collected through WeChat.
2. The results obtained in this study were the same as those in Hajli's research. Under this empirical approach, the model achieved good reliability, validity, and discriminant validity. $R^2$ had enough explanatory power. All the paths in the research model showed a positive effect.
3. From the comparison between Hajli's research and this study, we got the similar results. We can say that the social business intention model used by Hajli has been verified by a sample of British people and also Chinese people. Therefore it can be said that the model has cross-cultural interpretation ability.

**Table 6.** The comparison between Hajli's research and this study.

|  | Hajli's Research | This Study |
| --- | --- | --- |
| SNS | Facebook | WeChat |
| Data Collecting | Paper and online questionnaire | WeChat mobile questionnaire |
| Participants | 95% UKers | 100% Chinese |
| Usable questionnaires | 230 | 1277 |
| explanation power | Social Commerce Intention R square: 0.606 | Social Commerce Intention R square: 0.463 |
|  | Social Commerce Intention R square: 0.330 | Social Commerce Intention R square: 0.271 |
| path coefficients | H1: Social Support –> Social Commerce Intention 0.127   *support | H1: Social Support –> Social Commerce Intention 0.239   *support |
|  | H2: Social Commerce Constructs –> Social Commerce Intention 0.500   ***support | H2: Social Commerce Constructs –> Social Commerce Intention 0.480   ***support |
|  | H3: Social Commerce Constructs –> Social Support 0.479   ***support | H3: Social Commerce Constructs –> Social Support 0.521   ***support |

## 5. Conclusions and Suggestions

### 5.1. Implications for Research

Social networks such as MicroBlogs, WeChat, and QQ are witnessing rapid development in China. As the growth of traditional e-commerce is slowing down, social commerce featuring the combination of e-commerce and social platforms is making its next move. Individuals are no longer just consumers who receive information from the Internet. Instead, they can generate and share contents with others and conduct social interactions and spontaneous promotions among themselves. In this regard, consumers, instead of traditional e-commerce sellers, are becoming the main actors for promotion. Hajli's research adopted the theories of social commerce constructs and social support and proposed a social commerce intention model. Hajli investigated how British people received and shared information in buying products and services and how this consumer behaviors affect other consumers' buying intentions. This study adopts Hajli's research model to investigate Chinese peoplein the social commerce environment of China. The results of empirical evaluation by PLS-SEM show that social commerce constructs encourage communication and social interaction of consumers through the SNSs. Such social commerce constructs have positive effects on social support and social commerce intentions. The effect on social commerce intention is stronger than on social support. This result indicates that consumers' sharing behaviors and interactions will influence other participants' behaviors and purchasing intentions on SNSs.

Social commerce intention is the pre-stage of social commerce. The above analysis result shows that the reviews and comments of products or services given by consumers may affect other people's behaviors. Such interactive social platforms attract more people to communicate online. Consumers conduct research in forums and communities or refer to recommendation systems before purchasing. Such a social commerce construct not only provides consumers with information support, but also increases their emotional support due to peer purchases. Social support is constructed from information support and emotional support. Social interactions such as those seen in social commerce constructs affect participants' behavior, help obtain social support, enhance social commerce intentions, and affect their decisions in the purchase process.

Comparedto the results of Hajli's research, this study got similar results. We can say that despite for the cultural differences of western people on SNSs platforms, this model was still applicable when we conducted the survey in China's Internet environment and SNSs. Thus, it can be said that Hajli's model has cross-cultural interpretation ability. This study's conclusions are the same as that of the

model and show even stronger explanatory power. This can be explainedby the cultural values (called Guanxi) dominating Chinese people's behavior [25]. Guanxi can be based on many factors, including locality and dialect, fictive kinship, family, workplace, social clubs, and friendship. The rise of SNSs is strengthening Guanxi and social commerce in China [26].

Social network sites are becoming increasingly popular in China. People commonly transmit and exchange information through social network sites. In addition, in a real and interactive Internet environment, efforts should be made to encourage consumers to share personal experiences, use social platforms as carriers to maintain store-customer relationships through interactions, promote interaction among consumers, and guide consumers to voluntarily introduce such sites to other users through ratings, reviews, forums, communities, and recommendations. Thus, social commerce will be generated through the social commerce constructs and will usher in prosperous development.

### 5.2. Implications for Practice

Following in the footsteps of traditional e-commerce, social commerce has gradually flourished in recent years, but this new commerce model still lacks a tenable theoretical basis. This study mainly makes the following contributions through the supporting theories of social psychology and the application model of the Internet: (1) it verifies Hajli's theoretical model and makes empirical cross-cultural and cross-platform comparisons to strengthen the practicability of the theoretical model; (2) it demonstrates the impact of the social commerce constructs on social support and social commerce on China's social platforms; and(3) it provides a starting point for enterprises to develop social commerce and provides a direction for social commerce intention, which is conducive to the upgrading of e-commerce.

### 5.3. Limitations and Future Research

This study adopted WeChat as an SNS and data collection tool, since WeChat is the most widely used SNS in China. Although SNSs are considered to be better ways for data collection in the study of social commerce, we collected data on a specific SNS, which might still have its limitations. Social commerce is on the ascent. With the vigorous development of actual commerce, the demand for front-end research has become more urgent, such as that on generational differences or differences between urban and rural areas. Hence, future studies may focus on these topics.

**Author Contributions:** Investigation, C.-H.L.; Methodology, C.-H.L.; Supervision, C.-W.C.; Writing—original draft, C.-H.L.; Writing—review & editing, C.-W.C. All authors have read and agreed to the published version of the manuscript.

**Funding:** This research received no external funding.

**Conflicts of Interest:** The authors declare no conflict of interest.

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
