# Peer review of "An Empirical Study of Social Commerce Intention: An Example of China"

_information, doi:10.3390/info11020099_

Round 1
Reviewer 1 Report
First of all, I want to say to the authors that this work present interesting ideas that are potentially of interest to both academics and practitioners. And this is well designed, carefully conducted empirical research for the field of Information Journal.
I find this work interesting and honest. But I would like to suggest some ideas to further improve the manuscript. The included comments must be seen as recommendations to improve the quality of the presented work.
This paper seeks to study consumers' social commerce behaviour. I find this goal very ambitious. As the title of the paper. In fact, I would recommend modifying it and being more specific.
The introduction of the paper delves adequately into the specific research methodology and sample issues. The review of literature it seems so enough, but this field of research is very important and there are significant applied studies. The bibliography should be updated. We do not find any paper for the year 2019 and only one of the year 2018 and only one for 2017.
The research questions are not well developed. It would be necessary to clarify the objectives and argued. On the other hand, I consider the hypotheses well-argued and well-founded.
The methodology used can be deemed appropriate, and the authors’ presentation of the results is clear and concise, thus facilitating the reader’s understanding, but it would be necessary to deepen in the limitations of the chosen methodology, especially if the survey has been self-administered. Has there been any incentive?
An empirical study is developed (but is so local) and it is focused on the peculiarities on the one specific country, China. It would be interesting analyse if cultural differences can affect the results.
Finally, the conclusions could be improved and supplemented by some additional analysis of the data. The author should highlight the main contributions of this research and it would be interesting develop managerial implications, research limitations and future research lines.
Reviewer 2 Report
The paper addresses an interesting and fashionable topic and I appreciate the authors' intention to better understand the impact of social networks on consumer power. At the same time, I suggest the following in order to improve the paper:
Research questions and objectives should be included in the Introduction, as well as indications on which are is the contribution the paper makes to the research in the field The three research assumptions are defined around "significant impact", but the authors do not explain how they measure whether the impact is significant or not; I think a better definition of the assumptions would me made around the "positive-negative" impact The authors use a questionnaire to collect responses from social network users, but they do not mention whether the sample is representative for the Chinese population, which puts under question their results and the conclusions drawn based on these results The Results part needs to be re-written in order to make the reader better understand these results, also by making reference to other research and discussing the results compared to other similar results in the literature In the Conclusions part, the authors should stress the implications of their results and not just repeat what is already included in the Introduction part of the paperAuthor Response
Please refer to attachment

Round 2
Reviewer 2 Report
I congratulate the authors for proposing an improved version of their paper. Although they still need to correct some English language deficiencies, the paper is better compared to the previous version.